# Optimizing Data Flow in Binary Neural Networks

**DOI:** 10.3390/s24154780

**Published:** 2024-07-23

**Authors:** Lorenzo Vorabbi, Davide Maltoni, Stefano Santi

**Affiliations:** 1Datalogic Labs, Via San Vitalino 12, 40012 Bologna, BO, Italy; lorenzo.vorabbi@datalogic.com (L.V.); stefano.santi@datalogic.com (S.S.); 2Department of Computer Science and Engineering (DISI), University of Bologna, Cesena Campus, Via dell’ Università 50, 47521 Cesena, FC, Italy

**Keywords:** binary neural networks, efficient deep learning, quantized neural networks

## Abstract

Binary neural networks (BNNs) can substantially accelerate a neural network’s inference time by substituting its costly floating-point arithmetic with bit-wise operations. Nevertheless, state-of-the-art approaches reduce the efficiency of the data flow in the BNN layers by introducing intermediate conversions from 1 to 16/32 bits. We propose a novel training scheme, denoted as **BNN-Clip,** that can increase the parallelism and data flow of the BNN pipeline; specifically, we introduce a clipping block that reduces the data width from 32 bits to 8. Furthermore, we decrease the internal accumulator size of a binary layer, usually kept using 32 bits to prevent data overflow, with no accuracy loss. Moreover, we propose an optimization of the batch normalization layer that reduces latency and simplifies deployment. Finally, we present an optimized implementation of the binary direct convolution for ARM NEON instruction sets. Our experiments show a consistent inference latency speed-up (up to 1.3 and 2.4× compared to two state-of-the-art BNN frameworks) while reaching an accuracy comparable with state-of-the-art approaches on datasets like CIFAR-10, SVHN, and ImageNet.

## 1. Introduction

Over the past decade, deep neural networks (DNNs) have exhibited remarkable accuracy on numerous datasets such as ImageNet [1], surpassing traditional approaches and occasionally even human experts [2,3,4,5]. These advancements have been obtained by increasing the depth and complexity of the network, resulting in the extensive utilization of computational resources and memory bandwidth. Large DNN models execute efficiently on costly GPU-based machines, but their deployment on edge devices (i.e., small mobile or IoT systems), which are typically more resource-constrained, is substantially prevented. To address this issue, a range of techniques, including network quantization [6,7,8,9,10], network pruning [11,12], and efficient architecture design [13,14], have been introduced.

Recent research on quantization (e.g., [7,15,16,17]) has revealed that a DNN model can be quantized to 1 bit (denoted as binarization), resulting in a significant speed-up compared to the full-precision network. The memory demand of a binarized DNN (BNN) is greatly diminished in comparison to a DNN of equivalent structure, as a considerable portion of weights and activations can be expressed using 1 bit, usually $\left\{-1,+1\right\}$. Moreover, high-precision multiply-and-accumulate operations can be substituted with faster XNOR and popcount operations.

Nevertheless, the aggressive quantization can lead to an accuracy reduction in BNNs compared to their full-precision counterparts. Some researchers have demonstrated that the decrease in performance is often related to the gradient mismatch issue resulting from the non-differentiable binary activation function [16]. The non-differentiability of the quantization functions hinders gradient back-propagation through the quantization layer. Hence, prior studies have utilized a straight-through estimator (STE) for gradient approximation for the binarization operation [7,18].

Additionally, the binarization of weights and activations results in feature maps of reduced quality and capacity, requiring the adoption of a mix of binary and floating-point layers. However, every time a binary layer is placed before a floating-point one, the pipeline’s efficiency is undermined by input/output layer data type conversion. Moreover, the internal parallelism of a binary layer relies on the accumulator bitwidth, typically preserved at 32 bits to avoid overflow. We introduce several optimizations that enable the training of a BNN using an interlayer data width of 8 bits. While most previous research on BNNs focuses on enhancing overall network accuracy, our goal is to preserve the initial accuracy while improving the efficiency. Our contributions (graphically highlighted in Figure 1a,b) can be summarized as follows:A novel training scheme is proposed to enhance the data flow within the BNN pipeline (Section 3.1). In particular, we introduce a clipping block to decrease the data width from 32 to 8 bits, thus reducing the internal accumulator size.In Section 3.2, we propose an optimization of the batch normalization [19] layer when applied after a binary operation, which reduces latency and simplifies deployment.We optimize the binary direct convolution method for ARM instruction sets in Section 3.3.

**Figure 1 sensors-24-04780-f001:**
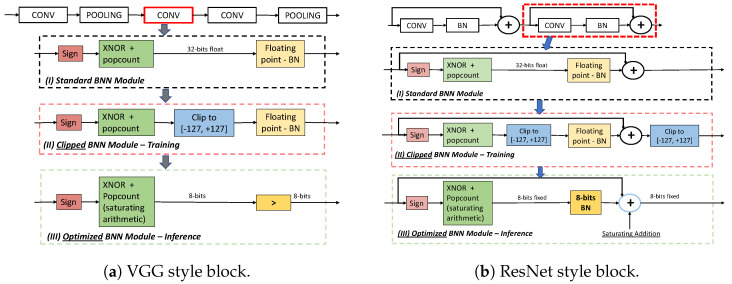
(*I*) Standard BNN blocks are used in [9,16]. (*II*) BNN block with output convolution clipping used during training. (*III*) Optimized BNN block adopted during inference. Popcount operation is performed using saturation arithmetic to keep the data width to 8 bits at inference time. BN is replaced by a comparison in case (**a**), while in (**b**), BN is 8-bit quantized.

To demonstrate the effectiveness of the proposed optimizations, as discussed in Section 4, we conduct experimental evaluations showcasing the speed-up compared to state-of-the-art BNN engines like LCE [20] and DaBNN [21] on datasets like CIFAR10, SVHN, and Imagenet.

## 2. Related Work

BNNs were initially introduced in [15], where the authors developed an end-to-end gradient back-propagation framework for training the binary weights and activations. While they achieved notable success on smaller classification datasets like CIFAR10 [22] and MNIST [23], they encountered a significant accuracy drop when applying the model to ImageNet. Several subsequent studies concentrated on enhancing BNN accuracy. For instance, the authors of [9] introduced XNOR-Net, which was further improved in [24]. In these approaches, real-valued scaling factors are utilized to scale the binary weight kernels. This methodology evolved to become a prominent binarization technique that aims to narrow the performance gap between BNNs and their real-valued counterparts. The Bi-Real Net [16] introduced shortcuts to efficiently propagate values across feature maps, leading to improved top-1 accuracy on ImageNet. However, the model continues to rely on 32-bit floating point for batch normalization and addition operations, as illustrated in Figure 1b.

One significant drawback of BNNs is the gradient approximation through the STE binarization function [15]. STE computes the derivative of the sign function as if the binary operator were linear, as shown in the following formula:(1)$$STE\left(x\right)=\left\{\begin{array}{cc} 1, & if\;x\geq -1\;and\;x\leq 1\\ 0 & \mathrm{otherwise}. \end{array}\right.$$
This current formulation of STE leads to gradient cancellation when inputs become too large [7]. This coarse approximation of the gradient with STE negatively impacts the testing accuracy of BNNs. To tackle this challenge, recent studies have explored various optimization techniques to enhance the performance of BNNs by refining the quantization process. Inspired by STE, numerous research works have devised methods to update the parameters approximately by introducing auxiliary loss functions; PCNN [25] proposes a projection convolutional network with a discrete back-propagation via projection. IR-Net [26] introduces a new parametrized binarization function to minimize both quantization error and information loss of parameters by balanced and standardized weights in forward propagation. RBNN [27] proposes a training-aware approximation of the sign function for gradient back-propagation. Similarly, AdamBNN [28] analyzes the influence of Adam and weight decay when training BNNs, showing that the regularization effect of second-order momentum in Adam is crucial to revitalizing dead weights. ReCU [29] proposes a rectified clamp unit to revive the dead weights, exploring the compromises between minimizing the quantization error and maximizing the information entropy.

To reduce the accuracy drop from the real-valued networks, some works propose the addition of a distribution loss or ad hoc regularization to the overall loss function. For instance, Real-to-Bin [17] incorporates an additional loss function that aligns spatial attention maps computed generated from binary and real-valued convolutions at the output. ReActNet [30] utilizes a distributional loss to encourage the binary network to learn output distributions similar to those of a real-valued network.

Recent research has seen the proposal of new network topology structures to enhance the BNN performance. High-capacity expert binary networks [31] focus on the information bottleneck of binary networks by introducing an effective expansion mechanism that maintains binary operations within the same budget. RepBNN [32] introduces a novel replaceable convolution module that improves feature maps by duplicating inputs or outputs along channel dimensions without additional parameter costs or convolution computations. PokeBNN [33] introduces a new efficient convolution block by adding multiple residual paths that are fused into a trainable activation function.

The quantization of weights and activations to 8 bits in neural networks is a well-established topic, as documented by Jacob et al. [34]. However, in BNNs, the implementation of 8-bit quantization is less prevalent, with the batch normalization (BN) layer typically executed using floating point arithmetic within standard inference engines [20,21]. In contrast, our research demonstrates that quantizing the BN layer and reducing the accumulator width within the binary operator can significantly accelerate the binary layer (comprising the binary operation and BN). Unlike previous studies [9,16,24,30], which apply binary convolution with scaling factors, our approach directly binarizes input activations and weights, subsequently quantizing the BN layer to eliminate the need for floating-point arithmetic. When the scaling factor is applied solely to the weights and the BN layer is placed immediately after the binary operation, the multiplication by the scaling factor can be integrated into the BN multiplication. However, the implementation of learnable biases (*ReAct Sign*), as suggested in ReactNet [30], during input binarization further increases the reliance on floating-point computation. Despite numerous efforts to develop more efficient and accurate architectures, only a limited number of studies have provided benchmarks on real devices, such as ARM processors. According to the analysis in Bannink et al. [20], the fastest inference engines for binary neural networks, with validated benchmarks (Section 4 of [20]), are LCE and DaBNN.

## 3. Data-Flow Optimizations

As illustrated in Figure 1a,b (*I*), the most commonly used binary neural network architectures, such as VGG and ResNet, contain four essential components within each convolution/fully connected (CONV/FC) layer: sign (binarization), XNOR, popcount, and batch normalization (BN). Given that both weights and inputs are binary, the conventional multiply-and-accumulate operation is substituted with XNOR and bit counting (i.e., popcount). To improve efficiency, XNOR and popcount are often combined. The adoption of batch normalization after each binarized layer is crucial in BNNs, as it significantly smooths the optimization landscape, resulting in more predictive and stable gradient behavior, thereby facilitating faster training. Figure 1a,b (*II* and *III*) illustrate our proposed optimizations for BNNs during training and inference. Before delving into these optimizations in detail, we will first identify the data-flow bottlenecks that impact existing solutions and then detail the methods we have developed to mitigate these issues.

In Figure 2a,b, we present an example of binary convolutional layer outputs for a VGG and a ResNet model. The ranges of activation values after the popcount operation (green histograms) exceed the interval $\left[-128;+127\right]$ (we actually consider the symmetric quantization interval $\left[-127;+127\right]$ because this choice enables a substantial optimization opportunity, as reported in Appendix B of [34]), indicating that adopting an 8-bit encoding would result in overflow. To prevent data loss, most existing BNN frameworks, including those discussed in [20,21], encode such data in 32-bit floating points. Conversely, the ranges of values after BN (red histograms in Figure 2) are more constrained.

In this paper, we propose accumulating the popcount output using 8-bit integers with saturation arithmetic through a two-stage training procedure designed to preserve model accuracy. In the subsequent subsection, we demonstrate the application of this technique to VGG and ResNet models.

### 3.1. Two-Stage Clipping

Our training procedure selectively executes or skips a clipping operation at each binary layer (row *b* of Figure 1a,b, blue blocks). To prevent accuracy loss when clipping is enabled, we introduce a two-stage training method. In the initial warm-up stage, the model is trained without any range constraints. In the subsequent stage (detailed in Algorithm 1), the network is trained with the clipping block enabled. This approach leverages the high accuracy achieved at the end of the first training stage, allowing the model to better tolerate 8-bit quantization during the second stage. Our experimental results indicate that this method preserves the accuracy of a model without clipping and, as discussed in Section 4.2, it can be easily adapted to other binarization methods.
**Algorithm 1:** Second-stage training procedure for BNNs.**Input:** The full-precision weights W; the input training dataset;**Output:** BNN model with convolution output clipped;Initialize network weights *W*;**repeat**    {*Forward Propagation*}    **for** l=1 **to** *L* **do**        Binarize floating point weights: $W_{bin}^{l}=sign\left(W^{l}\right)$;        Binarize floating point features of previous layer: $F_{bin}^{l-1}=sign\left(F^{l-1}\right)$;        Compute binary convolution: $F_{out}^{l}=F_{bin}^{l-1}*W_{bin}^{l}$;        Clip $F_{out}^{l}$ values to interval $\left[-127;+127\right]$ with: $F_{out\;clipped}^{l}=max\left(min\left(127,F_{out}^{l}\right),-127\right)$;        Execute Batch Normalization: $BN\left(F_{out\;clipped}^{l}\right)=\gamma^{l}\frac{F_{out\;clipped}^{l}-\mu^{l}}{\sigma^{l}}+\beta^{l}$;    **end for**    {*Backward Propagation*}    **for** l=1 **to** *L* **do**        Compute gradients based on the binarization weights $W_{bin}^{l}$, clipped convolutions $F_{out\;clipped}^{l}$ and batch normalization output $BN\left(F_{out\;clipped}^{l}\right)$;        Update full-precision weights $W^{l}$;    **end for****until** Convergence

### 3.2. Batch Normalization Optimization

The BN layer after the clipping is also optimized/8-bit quantized to increase the data flow of the inference pipeline. The batch normalization layer scales and shifts the output of the CONV/FC layer as follows:(2)$$BN\left(F_{out}^{l}\right)=\gamma \frac{F_{out}^{l}-\mu }{\sigma }+\beta$$
where γ,μ,σ and β are learned parameters and $F_{out}^{l}$ is the output feature of layer *l* that is the input of the BN function.

The BN optimization depends on the network model: VGG or ResNet. In both cases, we show that it is possible to keep the interlayer data type to 8-bit with appropriate changes to the binary layer structure.
**VGG style block**. When the BN layer is inserted in a pipeline similar to Figure 1a, where the following block is still binary, the BN operation can be simplified by replacing multiplication and division in Equation (Equation 2) with a simple comparison with a threshold τ. The simplification of Equation (Equation 2) leads to the following:
(3)$$sign\left(BN\left(F_{out}^{l}\right)\right)=\left\{\begin{array}{cc} +1 & \;\;if\;\;BN\left(F_{out}^{l}\right)\geq 0\\ -1 & otherwise \end{array}\right.\;\gamma \frac{F_{out}^{l}-\mu }{\sigma }+\beta \geq 0\Rightarrow \tau ≐\mu -\beta \frac{\sigma }{\gamma }\;sign\left(BN\left(F_{out}^{l}\right)\right)=\left\{\begin{array}{c} +1\;if\;F_{out}^{l}\geq \tau \;else\;-1\;\left(when\;\frac{\gamma }{\sigma }\geq 0\right)\\ -1\;if\;F_{out}^{l}\leq \tau \;else\;+1\;\left(when\;\frac{\gamma }{\sigma }<0\right) \end{array}\right\}$$
The threshold τ of Equation (Equation 3) can be computed offline and easily quantized to 8 bits to exploit the output features of layer *l*. Therefore, when multiple BNN modules are stacked, batch normalization can be replaced by a threshold comparison according to Equation (Equation 3). Even if BN can be replaced with a threshold comparison, 8-bit data flow is still important because it allows accumulating the binary xnor and popcount operations directly on 8 bits using saturation arithmetic instead of the standard 32 bits.**ResNet style block**. When a BNN block is placed in a ResNet style pipeline, followed by an addition operator, Figure 1b, the BN layer can be executed with both scaling and bias factors to 8 bits. As reported in Figure 3, the internal data representation of a quantized BN layer is expanded to 16 bits to preserve accuracy during quantization, but the input/output data type remains within 8 bits. The iterative uniform quantization procedure we adopted is symmetric and keeps the zero-point representation unaltered, as reported in Algorithm 2. The procedure iterates over the BN floating-point layers and, for each one, computes the quantization scale, quantized, freezes the weights, and retrains the remaining layers.

### 3.3. Binary Direct Convolution Optimization on ARM

General matrix multiplication (GEMM) is a widely adopted method for implementing efficient convolutions. However, as reported in [35], the GEMM approach increases the memory footprint of the model, complicating the porting of the model to embedded devices. Additionally, GEMM routines are not always optimized for convolutions on ARM devices, particularly ARMv8 (64-bit ARM architecture) and its relevant operations such as *vcount* and *addv*.

*vcount* takes an N-byte vector as input and outputs an N-byte vector containing the number of 1 s present in each input byte. *addv* takes an N-byte vector as input and outputs the sum of the N bytes as one single value.

Inspired by [21,35], we propose a hybrid direct binary convolution (see Figure 4) that utilizes both the *addv* instruction and common *add* operations. The binary convolution process comprises three distinct steps: binarization/bit-packing, padding, and convolution. While [21] executes these steps sequentially, we introduce a more cache-friendly approach that combines these steps into a single operation executed with tiling. Specifically, as detailed in the pseudo-code of Algorithm 3, the inner loop computes the output for multiple kernel positions (e.g., two positions in the example of Algorithm 3), while sign extraction and packing are performed only once for all channels. This method is supported by the analysis in Table 1, which demonstrates that processing multiple kernel positions reduces the instructions count in the inner loop (last column of Table 1). Additionally, we designed an alternative kernel memory layout that is more compatible with ARMv8 SIMD processing instructions, as illustrated in Figure 5.
**Algorithm 2:** Procedure to quantize the BN floating point layers in a BNN model where convolution output is clipped.**Input:** The full-precision weights W; the input training dataset;**Output:** BNN model with BN float layers replaced by 8-bit quantized version;**for** l=1
 **to** 
*L* **do**    **if** *l* is BN floating point **then**        Range of features $F_{out}^{l}$ as: $Range^{l}=\left[min\left(F_{out}^{l}\right);max\left(F_{out}^{l}\right)\right]$;        {$N^{l}$
*is the number of layer variables (4 for BN)*}        **for** h=1 **to** $N^{l}$ **do**           Range of weights $w_{h}^{l}$ as: $Range_{w_{h}^{l}}=\left[min\left(w_{h}^{l}\right);max\left(w_{h}^{l}\right)\right]$;           {*1 bit is reserved for sign*}           Bits used for integer part: $IntegerBits_{w_{h}^{l}}=clip\left(\left\lceil log_{2}\left(max\left(abs\left(Range_{w_{h}^{l}}\left[0\right]\right),\left(Range_{w_{h}^{l}}\left[1\right]\right)\right)\right)\right\rceil ,0,15\right)$;           Bits used for fractional part as: $FracBits_{w_{h}^{l}}=15-IntegerBits_{w_{h}^{l}}$;        **end for**        Integer bits for weights of layer *l*: $IntegerBits_{w^{l}}=max\left(IntegerBits_{w_{h}^{l}}\right)$;        Fractional bits for weights of layer *l*: $FracBits_{w^{l}}=15-IntegerBits_{w^{l}}$;         {*Add quantization noise to floating point weights*}        **for** h=1 **to** $l^{N}$ **do**           $w_{q_{h}}^{l}=\frac{1}{FracBits_{w^{l}}}round\left(2^{FracBits_{w^{l}}}*w_{h}^{l}\right)$;           Replace $w_{h}^{l}$ with $w_{q_{h}}^{l}$;        **end for**        Freeze $w^{l}$ weights and retrain the model;         {*Export the quantized weights of layer l for deployment*}        **for** h=1 **to** $l^{N}$ **do**           $w_{quantized_{h}}^{l}=round\left(2^{FracBits_{w^{l}}}*w_{q_{h}}^{l}\right)$;        **end for**    **end if****end for**

The implementation details of our binary convolution are illustrated in Figure 4. The operation *Extract sign bit* performs binarization, bit-packing, and padding. Subsequently, the (bit-wise) XNOR output is processed by the popcount operation, which includes *vcnt 8-bit wise*, *add*, and *addv*. On the ARM architecture, this can be implemented using the *vcount* instruction along with a sequence of *addv* instructions. To optimize performance, we implement several pair-wise additions followed by a final *addv* instruction, as the *addv* instruction is more computationally expensive. The convolution process operates without intermediate outputs, processing the input data in its entirety. Notably, the clipping operation can be efficiently executed on ARM devices by leveraging saturation arithmetic. By prefixing the addition operations (*add* and *addv*) with *q* (indicating saturation arithmetic), we can confine the results to the fixed range $\left[-127;+127\right]$ and use the max function to avoid the −128 value.
**Algorithm 3:** Binary convolution pseudo-code for input tensor of size 28×28×128 and kernel size 3×3×128.**Input:** The input tensor (i.e., 28×28×128) *IT*;**Output:** Output of binary convolution;{*Loop on rows*}**for** j=0 **to** 28 **do**     {*Loop on columns. To exploit data locality, two kernel positions are computed}*    **for** i=0 **to** 28 **do**        Extract sign bit (storing on the stack) of two input patches for all 128 channels,         $\begin{array}{c} BIT\left(j,i\right)=Binarize\left(IT\left(j,i\right)\right)\\ BIT\left(j,i+1\right)=Binarize\left(IT\left(j,i+1\right)\right) \end{array}$;         {*Loop on channels*}        **for** t=0 **to** 128 **do**           Load packed binary weight tensors (WT) of 2 channels: $WT\left(t,t+1\right)$;           Execute XNOR operation: $XNOR\left(BIT,WT\right)$;           Accumulate using the popcount (*vcntq*);           Finalize the accumulation using the *vaddq* and *vaddvq* (as last step) instructions. The accumulator is 8-bit wide using saturation arithmetic;           Store final output;           t←t+2;        **end for**        i←i+2;    **end for**    j←j+1;**end for**

## 4. Experimental Results

In this section, we first evaluate the efficiency result of BNN-Clip compared to the state-of-art BNN frameworks such as LCE and DaBNN; the comparison is carried out on real hardware devices like Raspberry Pi Model 3B and 4B with 64-bit OS, which better represent a low-power platform compared to HUAWEI Kirin and Apple M1, benchmarked in [36]. Then, we present various accuracy benchmarks of the proposed two-stage training procedure, focusing on CIFAR-10, SVHN, and ImageNet, and several architectures: VGG, Resnet-18, Resnet-50, and ReActnet-A.

### 4.1. Efficiency Analysis

To validate the efficiency of our method, we first focused on the convolution macro-block (extending the results reported in [20] to include the Raspberry Pi 3) depicted in Figure 1. We compared the efficiency of the BNN-Clip approach with LCE and DaBNN, which, to the best of our knowledge, are the fastest inference engines for binary neural networks. The kernel dimensions were selected to match those used in the *BiRealnet* and *ReActnet* architectures. Our assessment was conducted on ARMv8 platforms, specifically the Raspberry Pi 3B and 4B. We implemented the convolution operation using ARM NEON *intrinsics* instead of inline assembly. Intrinsics allow for code that is easier to maintain and can automatically adapt to both ARMv7 and ARMv8 platforms without a significant performance loss compared to pure assembly code. In Figure 6, we present the comparison results for the Rpi 3B and 4B targets. Our solution demonstrates a clear performance improvement for single binarized convolutions across all kernel sizes. Including all the optimizations introduced in Section 3, our method accelerates binary convolution by up to 1.32 and 2.4× compared to LCE and DaBNN, respectively, with an average improvement of 1.2 and 1.71×.

Table 2 and Table 3 (Raspberry Pi 3B and 4B, respectively) compare the measurements of the overall latencies of several BNN architectures, *BiRealnet*, *ReActnet*, *Quicknet*, and *QuicknetLarge*, using LCE and daBNN frameworks. Our solution uses LCE but we replaced the original binary convolution implementation with our optimized one. As reported in the last column of the tables, daBNN is significantly slower compared to LCE and our method on both RPi targets. Instead, our optimized binary convolution constantly outperforms LCE in inference time for all the models considered, reaching up to 1.3× maximum speed-up. In particular, our solution reaches an average speed-up of 1.19 and 1.16× on Raspberry Pi 3B and 4B.

### 4.2. Accuracy Analysis

We evaluated two VGG-style networks (VGG-11 and VGG-Small) and a ResNet-18 architecture for CIFAR-10 and SVHN, applying several binarization methods. VGG-11 [37] and VGG-Small [38] are both high-capacity networks for classification. To validate the advantages of our solution on a more realistic and challenging dataset like Imagenet, we applied our methodology to several state-of-the-art binary neural networks with publicly available source code.

**Results on CIFAR10 and SVHN**. For CIFAR10, the RGB images were scaled to the interval $\left[-1.0;+1.0\right]$, and the following data augmentation was used: zero padding of 4 pixels for each side, a random 32×32 crop, and a random horizontal flip. No augmentation was used at test time. The models were trained for 140 epochs.

On SVHN, the input images were scaled to the interval $\left[-1.0;+1.0\right]$ and the following data augmentation procedure was used: random rotation (±8 degrees), zoom ($\left[0.95,1.05\right]$), random shift ($\left[0;10\right]$), and random shear ($\left[0;0.15\right]$). The models were trained for 70 epochs.

All the networks were trained using the same training procedure without adopting additional distillation losses, to improve the accuracy of BNN models.

The accuracy achieved by the models is reported in Table 4, showing that the clip operation does not substantially affect the overall accuracy, and the two-stage clipping allows the preservation of the original accuracy. Figure 7 and Figure 8 show the training and validation curves on CIFAR10 and SVHN; we can note that a limited number of epochs is necessary during the second training stage to recover accuracy.

**Results on ImageNet**. On Imagenet, we considered the following architectures: BinaryResNetE18 and BinaryDenseNet28 [42], BiRealnet [16], ReActnet [30], ReCU [29], ReActnet-AdamBNN [28], RepAdamBNN [32], and PokeBNN [33]. For all the previous networks, we used pretrained models, available online, and we executed only the second training step by modifying, according to Figure 1b, the BNN architecture to insert the clipping operation within the back-propagation framework. Table 5 displays the performance comparison of the pretrained model without (column *top1(Raw)*) and with the clipping block (column *top1(BNN-Clip)*). In Table 5, we can see that our solution marginally affects the model accuracy. Specifically, in 80% of cases, the accuracy drop is less than 2%, and for BinaryResNetE18 and BinaryDenseNet28 models, our solution completely recovers the original accuracy. Overall, on average, the accuracy drop is around 1%, representing a good compromise compared to an average speed-up of 1.2×, as reported in Table 2 and Table 3.

## 5. Conclusions

This paper introduced several optimizations in the BNN data flow that, together, achieve a maximum speed-up of 1.3 and 2.4× compared to state-of-the-art BNNs frameworks, LCE and daBNN, without any accuracy loss for at least one full-precision model. We first analyzed how to speed up the binary convolution by reducing the bitwidth (8 bits) of the internal accumulator. Then, we proposed a double-stage training step that allows the preservation of the accuracy model when the bitwidth restriction is applied. In addition, we showed how to speed up the binary convolution, proposing an *ad hoc* processing pipeline for the ARM NEON instruction set. Experimental results demonstrate that our solution can be applied to state-of-the-art binarization techniques, significantly reducing the model latency (up to 2.4×) with a minimal (1%) or no accuracy drop. In the future, we intend to investigate the simplification of the training procedure, possibly collapsing it into a single stage to reduce training time and complexity.

## Figures and Tables

**Figure 2 sensors-24-04780-f002:**
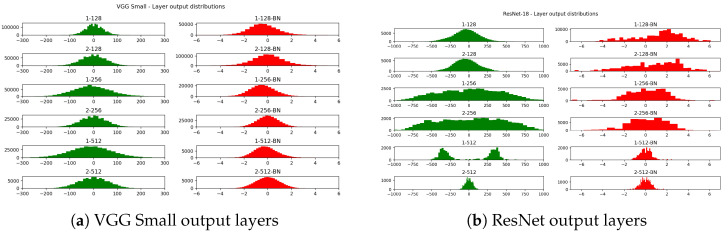
Example of the output distributions after binary convolution. Panel (**a**) refers to a VGG style network while (**b**) refers to a ResNet architecture. Green shows the distribution before the BN layer, and red shows afterward.

**Figure 3 sensors-24-04780-f003:**
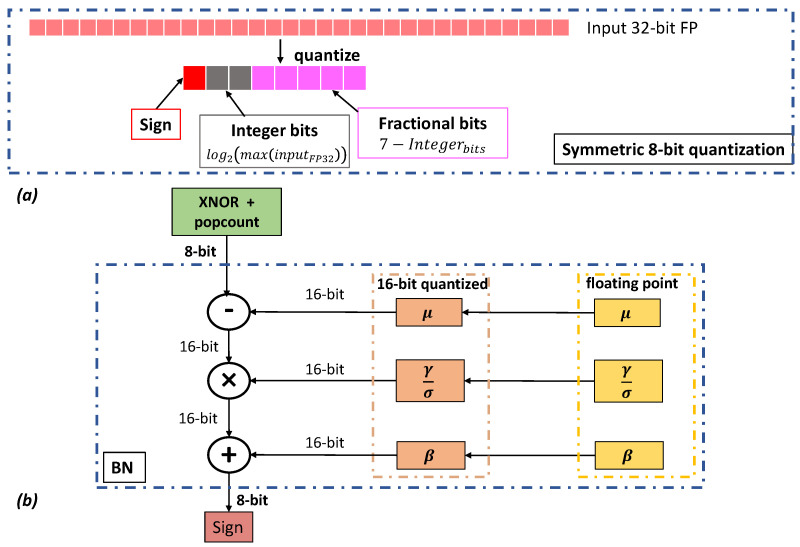
(**a**) 8-bit symmetric quantization procedure that reserves fractional/integer bits based on the range of input 32-bit floating point values. (**b**) Implementation of the BN layer with 8-bit quantization using an internal 16-bit representation to preserve accuracy.

**Figure 4 sensors-24-04780-f004:**
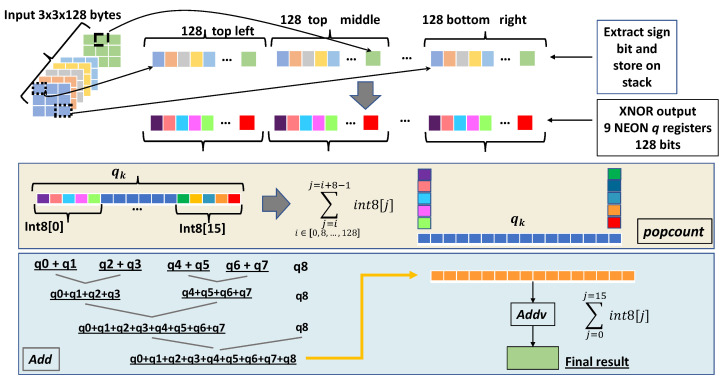
The 3×3×128 input patch is convolved (XNOR + popcount) with one kernel through the extract sign bit, XNOR, and then popcount operations. Popcount is performed using *vcnt*, summing in pairs the *vcnt* output, and the last step uses the *addv* operation.

**Figure 5 sensors-24-04780-f005:**
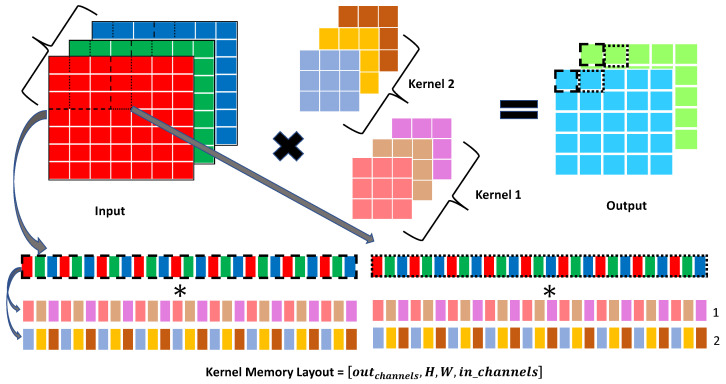
The 7×7 input image with 3 different channels (denoted by color) is convolved with two separate kernels to obtain a 5×5 output with two output channels. To better exploit the SIMD 128-bit registers, a different memory layout for kernel is devised: $\left[out_{channels},\;H_{filter},\;W_{filter},\;in_{channels}\right]$.

**Figure 6 sensors-24-04780-f006:**
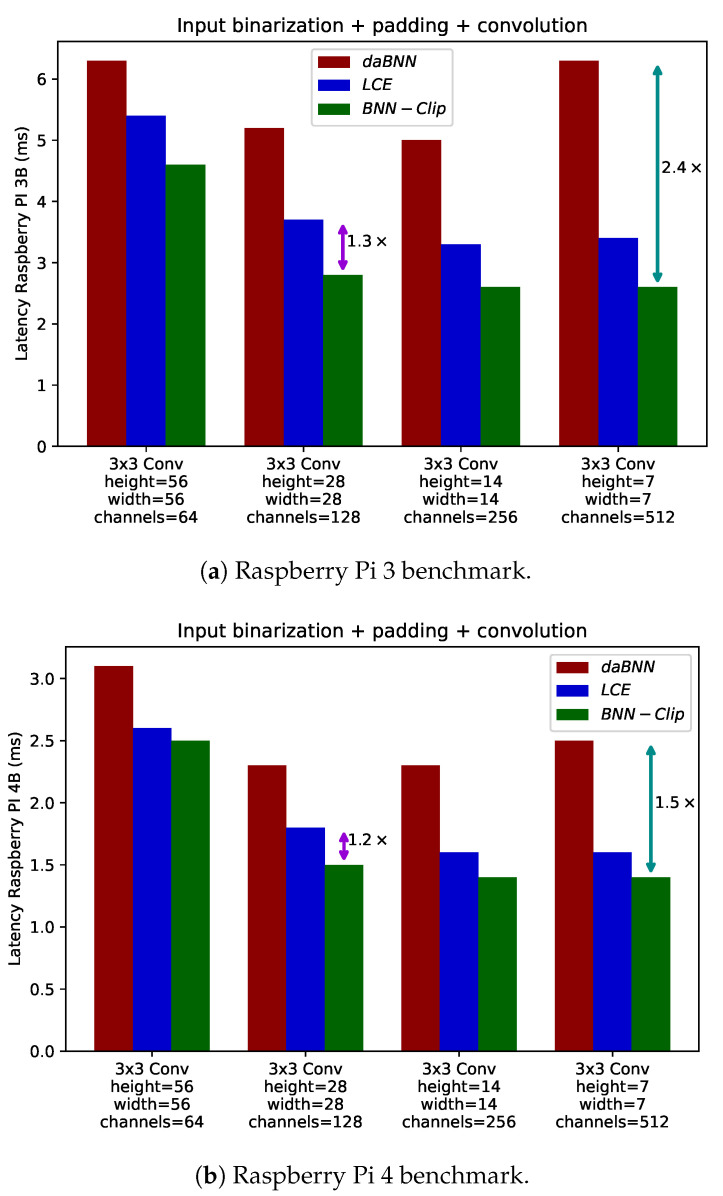
Latency evaluation of our method compared to DaBNN and LCE on Raspberry Pi 3B (**a**) and 4B (**b**) devices for four different combinations of input and kernel sizes.

**Figure 7 sensors-24-04780-f007:**
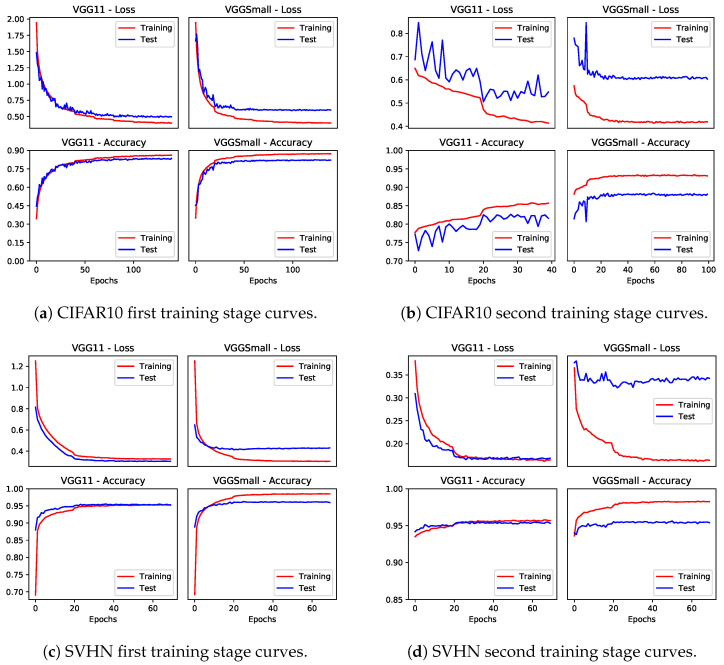
Training loss and testing accuracy curves for VGG11 and VGGSmall on CIFAR10 and SVHN of the first and second training stages.

**Figure 8 sensors-24-04780-f008:**
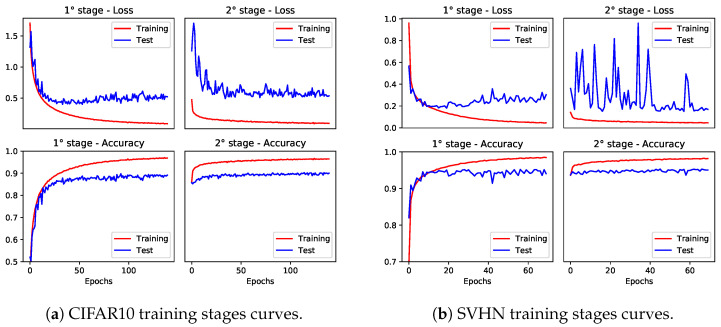
Training loss and testing accuracy curves for ResNet-18 on CIFAR10 and SVHN for both training stages.

**Table 1 sensors-24-04780-t001:** Analysis of the computational cost of performing the binary convolution as reported in the inner loop (Line 6) of Algorithm 3. In particular, two kernel positions (for the example considered) are computed within the inner loop (last row). The last column shows that by increasing the number of kernel positions, it is possible to reduce the instructions count.

Precision	# XNOR	# Popcount	# *vaddq*	# *vaddvq*	$\frac{\#\mathrm{instruction}}{\#\mathrm{output}\;\mathrm{values}}$
*binary (1 kernel pos)*	9	9	8	1	21
*binary (2 kernel pos)*	12	12	11	2	18.5
*binary (3 kernel pos)*	15	15	14	3	14

**Table 2 sensors-24-04780-t002:** Comparison of the full model latencies of our solution compared to LCE and daBNN measured in milliseconds on Raspberry Pi 3B.

Architecture	*daBNN*	*LCE*	*BNN-Clip*	Speed-Up *BNN-Clip* vs. *LCE*
*BiRealnet*	240	157	144	**1.09**
*ReActnet*	unavailable	298	248	**1.2**
*Quicknet*	unavailable	97	80.8	**1.2**
*QuicknetLarge*	unavailable	167	130	**1.28**

**Table 3 sensors-24-04780-t003:** Comparison of the full model latencies of our solution compared to LCE and daBNN measured in milliseconds on Raspberry Pi 4B.

Architecture	*daBNN*	*LCE*	*BNN-Clip*	Speed-Up *BNN-Clip* vs. *LCE*
*BiRealnet*	120	81.6	75.5	**1.08**
*ReActnet*	unavailable	156	130	**1.2**
*Quicknet*	unavailable	49	41.5	**1.18**
*QuicknetLarge*	unavailable	84	69.7	**1.2**

**Table 4 sensors-24-04780-t004:** Accuracy comparison (top-1) of our method with SOTA on CIFAR10 and SVHN.

Method	Topology	Bitwidth	CIFAR10 %	SVHN %
BNN [15]	VGGSmall [38]	32 FP	93.8	96.5
Main/Subs. Net.	VGG11 [37]	32 FP	83.8	-
ResNet-18 [26]	ResNet-18	32 FP	93.0	97.3
BNN	VGGSmall	1-bit	89.9	96.5
XNOR-Net [9]	VGGSmall	1-bit	82.0	96.5
Bop [39]	VGGSmall	1-bit	91.3	-
BNN-DL [40]	VGGSmall	1-bit	89.9	97.2
IR-Net [41]	VGGSmall	1-bit	90.4	-
Main/Subs. Net.	VGG11	1-bit	82.0	-
BiRealnet [16]	ResNet-18	1-bit	89.3	94.7
ReActNet [30]	ReActnet-A	1-bit	91.5	95.7
**BNN-Clip**	VGGSmall	1-bit	88.8	96.1
**BNN-Clip**	VGG11	1-bit	**83.7**	95.5
**BNN-Clip**	ResNet-18	1-bit	90.3	95.3

**Table 5 sensors-24-04780-t005:** Accuracy comparison of our method with SOTA on ImageNet. Top1(Raw) reports the results achieved by original methods, while top1(BNN-Clip) shows the results obtained by applying the techniques detailed in Section 3. The last column shows the accuracy drop of our binary convolution clipping when applied to the state-of-the-art methods.

Method	Topology	top1(Raw) %	top1(BNN-Clip) %	Δ(%)
XNOR-Net [9]	ResNet-18	51.2	51.1	**−0.1**
BiRealnet [16]	ResNet-18	56.4	56.1	−0.3
BinaryResNetE18 [42]	ResNet-18	58.1	58.1	**0.0**
BinaryDenseNet28 [42]	DenseNet-28	60.7	60.7	**0.0**
ReActnet [30]	ReActnet-A	68.2	66.4	−1.8
ReCU [29]	ResNet-18	61.0	58.0	−3.0
ReCU [29] with training settings of [30]	ResNet-18	66.4	65.4	−1.0
AdamBNN [28]	ReActnet-A	70.2	69.0	−1.2
RepAdamBNN [32]	ReActnet-A	71.5	69.3	−2.2
PokeBNN-0.75x [33]	ResNet-50	70.5	69.2	−1.3

## Data Availability

Evaluation of the solution was performed using the CIFAR10, SVHN, and Imagenet datasets.

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
