# Peer review of "Optimizing Data Flow in Binary Neural Networks"

_sensors, 2024, doi:10.3390/s24154780_

Round 1
Reviewer 1 Report
Comments and Suggestions for Authors
This manuscript introduced several optimizations to speed up the data-flow in Binary Neural Networks. Experimental results show the performance of the proposed schemes. The topic is interesting and have good application prospect. The work is solid, and the manuscript is well organized. Some small flaws:
(1) In the simulations, there is no description about the results in Figure 7 & 8. The authors can add some description for them.
(2) The are some typos in the manuscript. For example, “*ouput* feature” in Line 156.
Author Response
Comments 1: In the simulations, there is no description about the results in Figure 7 & 8. The authors can add some description for them.
Response 1: The details and the description of the experiments results reported in Figures 7 and 8 are in Section 3.2 which reports: normalization, augmentation pipeline and epochs.
Comments 2: The are some typos in the manuscript. For example, “*ouput* feature” in Line 156.
Response 2: Agree. We have, accordingly, fixed the typo.
Reviewer 2 Report
Comments and Suggestions for Authors
This paper suggests a few approaches to optimize the data flow in the binary neural networks. The paper is well structured and easy to follow. However I have a few remarks that should be addressed to improve the paper.
First of all, you have cited your own work with the exact same title [22]. What are the differences? I wasn't able to find that exact paper, but I found similar ones sent to other conferences and on arxiv.
Obtained results demonstrate that your approach works comparing the state-of-the art methods. But I wonder whether it's so because of the proper implementation rather than theoretical improvements? The overall ideas are not new, so I wonder if the result has something to do with the studied architectures?
I'd suggest avoiding using 'ours' in figures and tables. A well-structured method or an approach deserves a name at least in the scope of one particular work.
In the description you have 'Fig 1. c)' but there is no part c.
Author Response
Comments 1: First of all, you have cited your own work with the exact same title [22]. What are the differences? I wasn't able to find that exact paper, but I found similar ones sent to other conferences and on arxiv.
Response 1: Agree. We have removed the citation as it is not relevant
Comments 2: Obtained results demonstrate that your approach works comparing the state-of-the art methods. But I wonder whether it's so because of the proper implementation rather than theoretical improvements? The overall ideas are not new, so I wonder if the result has something to do with the studied architectures?
Response 2: Thank you for pointing this out. The speed-up obtained on ARM architectures was possible because we introduced the clipping block that allows to accumulate the output of the popcount over 8-bit quantized values. This result allowed to quantize also the BN layer removing completely the floating-point computation from the BNN module. This result is valid also on other architectures such as x86 or MIPS but the speed-up achievable is always architecture dependent as many factors (instruction set, clock frequency, memory controller) contribute to the final result.
Comments 3: I'd suggest avoiding using 'ours' in figures and tables. A well-structured method or an approach deserves a name at least in the scope of one particular work.
Response 3: We agree with this comment and we’ll change with an appropriate signature.
Comments 4: In the description you have 'Fig 1. c)' but there is no part c.
Response 4: Thank you for pointing this out, Figure 1 is composed by 2 sub-figures: a and b. Each sub-figure is split in 3 parts, named respectively a), b) and c). The c) symbols refers to one these 3 parts. We agree on this and the symbols used are not understandable and we changed it using I), II) and III).
Reviewer 3 Report
Comments and Suggestions for Authors
The article focuses on enhancing the efficiency and performance of Binary Neural Networks (BNNs) through innovative optimization techniques. The authors introduce a novel training scheme that includes a clipping block to reduce data width and internal accumulator size in BNN layers. By optimizing the Batch Normalization layer and implementing Binary Direct Convolution for ARM NEON instruction sets, the study aims to improve data flow and parallelism in BNN pipelines. Experimental results demonstrate significant speed-up in inference latency compared to state-of-the-art BNN frameworks while maintaining accuracy levels on datasets like CIFAR-10, SVHN, and ImageNet. Overall, the research highlights the importance of optimizing data flow in BNNs to achieve efficient deep learning and quantized neural networks. The major issues of the article are listed:
1.- Provide more detailed explanations of the specific algorithms or techniques used for each optimization could enhance the clarity of the methods section.
2.- Add visual aids such as tables or graphs to illustrate the performance improvements achieved through the proposed optimizations could enhance the clarity and impact of the results section.
3.- Provide a more detailed explanation of the novel training scheme introduced, including the specific steps involved in implementing the clipping block and how it contributes to improving data flow in BNN pipelines. This will help readers better understand the core methodology of the study.
4.- Expand on the optimization of the Batch Normalization layer post-binary operation. Describe how this optimization reduces latency and simplifies deployment, highlighting the technical aspects of the enhancement and its impact on overall performance.
5.- Offer a comprehensive explanation of the optimized implementation of Binary Direct Convolution for ARM NEON instruction sets. Detail the modifications made to leverage ARM instruction sets effectively and how this optimization enhances the efficiency of BNN layers.
6.- Incorporate algorithmic pseudocode or flowcharts illustrating the key processes involved in the novel training scheme, Batch Normalization optimization, and Binary Direct Convolution implementation. Visual aids can enhance the clarity of the methods section and facilitate understanding for readers.
7.- Conduct a detailed comparative analysis with existing BNN frameworks like LCE and DaBNN to showcase the specific performance improvements achieved through the proposed optimizations. Include quantitative metrics and benchmarks to support the claims of speed-up and accuracy levels.
8.- Address the potential trade-offs associated with reducing data width from 32 bits to 8 bits in the clipping block. Discuss how this reduction impacts accuracy, computational efficiency, and any potential challenges that arise from narrowing the data width in BNN layers.
9.- Explain the significance of reducing the internal accumulator size in a binary layer while maintaining accuracy. Detail how this optimization contributes to improving data flow, reducing resource consumption, and enhancing the overall efficiency of BNN pipelines.
10.- Conclude the article by outlining potential future research directions stemming from the proposed optimizations. Discuss areas for further exploration, such as scalability to larger datasets, extension to different neural network architectures, or integration with emerging technologies, to inspire continued advancements in optimizing data flow in BNNs.
Author Response
Comments 1: Provide more detailed explanations of the specific algorithms or techniques used for each optimization could enhance the clarity of the methods section.
Response 1: This comment is very generic, and we are not sure about the “more detailed explanations” the reviewer is pointing out. Algorithms 1, 2 and 3 deeply cover all the optimizations introduced and the section 2.3 highlights all the optimization performed on ARM NEON
Comments 2: Add visual aids such as tables or graphs to illustrate the performance improvements achieved through the proposed optimizations could enhance the clarity and impact of the results section.
Response 2: In our analysis we reported several Tables/Figures to show the advantages of our approach: Table 1 shows the theoretical foundation of our approach that can accelerate binary convolution. Figures 6 shows the speed-up of our method by considering a single binary conv block. Tables 3 and 4 reports the speed-up of the overall pipeline using our solution.
Comments 3: Provide a more detailed explanation of the novel training scheme introduced, including the specific steps involved in implementing the clipping block and how it contributes to improving data flow in BNN pipelines. This will help readers better understand the core methodology of the study.
Response 3: Algorithms 1 and 3 detail all the necessary steps to perform the double training stages showing how to implement the clipping operation. Figure 1 shows that the clipping operation can be implemented for free by using the saturating arithmetic.
Comments 4: Expand on the optimization of the Batch Normalization layer post-binary operation. Describe how this optimization reduces latency and simplifies deployment, highlighting the technical aspects of the enhancement and its impact on overall performance.
Response 4: The optimization of BN reduces latency as it is quantized compared to the original floating-point version and it simplifies the deployment as it can work directly with 8-bit input values which are the result of the accumulation of the binary convolution. The standard approach accumulates the popcount result over 32-bit and if we would reuse our BN optimized version we should reduce the bit-width of the input data resulting in a sub-optimal implementation.
Comments 5: Offer a comprehensive explanation of the optimized implementation of Binary Direct Convolution for ARM NEON instruction sets. Detail the modifications made to leverage ARM instruction sets effectively and how this optimization enhances the efficiency of BNN layers.
Response 5: The description of the Binary Direct Convolution is detailed in section 2.3 where we specify that the usage of the NEON instruction addv is limited only to the last instruction as this operation requires more clock cycles compared to the standard add instruction. This solution allows to speed up the binary conv block.
Comments 6: Incorporate algorithmic pseudocode or flowcharts illustrating the key processes involved in the novel training scheme, Batch Normalization optimization, and Binary Direct Convolution implementation. Visual aids can enhance the clarity of the methods section and facilitate understanding for readers.
Response 6: We do believe this comment is overlapping with comments 3 and 4 that we already addressed.
Comments 7: Conduct a detailed comparative analysis with existing BNN frameworks like LCE and DaBNN to showcase the specific performance improvements achieved through the proposed optimizations. Include quantitative metrics and benchmarks to support the claims of speed-up and accuracy levels.
Response 7: The primary quantitative metrics are the latency and the memory usage. Our solution uses less memory as we do not need 32-bit accumulators and the latency comparison is reported in figure 6 (for single binary conv) and in tables 2 and 3 to consider the entire pipeline.
Comments 8: Address the potential trade-offs associated with reducing data width from 32 bits to 8 bits in the clipping block. Discuss how this reduction impacts accuracy, computational efficiency, and any potential challenges that arise from narrowing the data width in BNN layers.
Response 8: This comment seems to overlap with Comments 9
Comments 9: Explain the significance of reducing the internal accumulator size in a binary layer while maintaining accuracy. Detail how this optimization contributes to improving data flow, reducing resource consumption, and enhancing the overall efficiency of BNN pipelines.
Response 8: Instead of using 32-bit as accumulator we employ 8-bit registers. By adopting this optimization, the model would get an impressive accuracy drop because, as showed in Figure 2, too much information would be drop. The second stage training scheme is aimed to force the network to compress information into the 8-bit range by using the clipping block. This solution allows to replace 32-bit accumulators with 8-bit ones reducing the necessary instructions in the binary convolution, therefore reducing the latency.
Comments 10: Conclude the article by outlining potential future research directions stemming from the proposed optimizations. Discuss areas for further exploration, such as scalability to larger datasets, extension to different neural network architectures, or integration with emerging technologies, to inspire continued advancements in optimizing data flow in BNNs.
Response 10: As reported in Section 10, the next future work is to fuse the into 1 step the training phase reducing the training time.
Round 2
Reviewer 2 Report
Comments and Suggestions for Authors
Thank you for addressing my questions. I'm a bit confused that there are no changes with regard to my Comment 3. In tables and figures you still use 'ours' though in your reply you've mentioned that it was changed.
Author Response
Comments 1: Thank you for addressing my questions. I'm a bit confused that there are no changes with regard to my Comment 3. In tables and figures you still use 'ours' though in your reply you've mentioned that it was changed.
Response 1: Agree. We have removed ours and added the method "BNN-Clip"
Reviewer 3 Report
Comments and Suggestions for Authors
The authors addressed all my concerns. Reviewed the plagiarism informe, it shows a 25% of similarity, in this state the authors must correct this issue.
Author Response
Comments 1: The authors addressed all my concerns. Reviewed the plagiarism informe, it shows a 25% of similarity, in this state the authors must correct this issue.
Response 1: Thank you for pointing out this. We have rephrased the paper and we do believe to have removed the similarity.